# Analysis of Quality Teaching and Learning from Perspective of University Students

**Marek Vaclavik** [1,2,*] **, Martin Tomasek** [3] **, Iva Cervenkova** [1,2] **and Barbara Baarova** [4]

1   Department of Education and Adult Education, Faculty of Education, University of Ostrava, Fr. Sramka 3, 70900 Ostrava, Czech Republic
2   Centre for Educational Research, Faculty of Education, University of Ostrava, Fr. Sramka 3, 70900 Ostrava, Czech Republic
3   Department of Czech Literature and Literature Criticism, Faculty of Arts, University of Ostrava, Realni 5, 70103 Ostrava, Czech Republic
4   Department of Human Geography and Regional Development, Faculty of Science, University of Ostrava, 30. Dubna 22, 70103 Ostrava, Czech Republic
*   Correspondence: marek.vaclavik@osu.cz

**Abstract:** This paper presents the results of empirical research focused on the quality of teaching and learning methods, from the perspective of master's students at one of the Czech universities. The research focused on learning outcomes, teaching forms and methods, and the use of ICT technologies following a quantitative survey in this area, which showed the need to examine the topic in depth and in a broader context. Data for the qualitative research were collected through in-depth interviews; the primary research method was focus groups. The data were processed and analysed by coding techniques. The results showed that students prefer teaching and learning outcomes associated with the use in future practice. The teaching forms depend on the teacher's style rather than on the declared description in the curriculum. Contrary to most practices, students prefer teaching methods that lead to active learning. The advantages are identified in the frame of involvement of ICT in teaching, which makes sense and positively impacts students' learning; however, the effect depends on how the teaching forms are used.

**Keywords:** university teaching; teaching and learning strategies; teaching quality; assessment; evaluations; educational media; ICT technologies

## 1. Introduction

Student evaluations are one of the paramount feedback procedures helping to improve the quality of teaching at universities. As is the case in other areas, the topic (here, concretely, the curriculum) has been changing. Particularly, the curriculum model is based on the preparation of future teachers. This approach is usually considered as significant in the frame of the students' evaluations, e.g., [1], which usually take place at the end of the semester and are based on the results of questionnaire surveys or interviews with students. Students' comments on individual training courses' strengths and weaknesses help design visions for improvement. With regards to the visions of the students' feedback, the possibilities of improving could be necessary to research, e.g., including technical-based innovations [2] or analyses of the utilization of essential study materials [3]. Teaching-quality assessment questionnaires are not new in tertiary education but have been used for more than 80 years, as older studies have shown [4]. In recent years, however, interviewing students about teaching has become increasingly important and has become part of the common practice of universities around the world [5,6].

According to a consensus in the higher education environment, student feedback improves teaching and helps develop effective teaching strategies. Alternative methods for developing effective teaching strategies in the additional sense can be seen, e.g., in [1,2],

as challenges within learning or including virtual reality (ICT methods). Aditomo and Kohler [7] state that teachers and their teaching strategies are the main factors that determine students' learning outcomes. Interesting research on the evaluation of higher education was published by Chou, Luo, and Ramser [8]. Based on students' comments, the authors attempted to identify the elements indicating good teaching. Subsequently, they observed how the emotional mood of students affects the perception of what is quality and poor quality teaching. Sembiring even argues that students' satisfaction or dissatisfaction is directly related to their learning successes or failures [9]. Other authors point out that academics need to attach importance to what interviewed students think of the quality of teaching, to understand the quality of teaching [10,11].

Several authors point out that the collected data are not used effectively by the university [4,5,12]. Borch et al. [4] explain the low level of use of the obtained data as showing a certain rigidity of academics' teaching; academics' fear of change, busyness, and questioning the relevance of students' statements; or simply pedagogical scepticism. Boring, Ottoboni, and Stark [13] call for caution when handling data. Drawing from an in-depth comparative study of two qualitatively different universities in the USA, they found that a student evaluation questionnaire was tied to gender biases rather than capturing the effectiveness of teaching itself. They also pointed out that, in the USA, the results of student questionnaires led to changes in teaching strategies and influenced personnel policy. Finally, they stated that many questionnaires used only estimated student satisfaction and student views on teaching, regardless of the learning outcomes.

Universities also have to deal with the low validity of results caused by the limited participation of engaged respondents or subsequent inaccurate work with the obtained data. At the same time, only the cooperation of all teaching actors and relevant analytical tools allows for identifying the factors that decide the improvement of teaching.

Nale [14] mentions another weakness in the quality of higher education studies. They state that ongoing research always focuses on only one level of quality—either on the performance component or on the importance of the factors influencing the quality of teaching. Thus, they recommend combining both levels in research [14]. Other authors [15,16] also support multifactor evaluation. They stress the importance of finding out what students expect in terms of quality at the university and asking them about their satisfaction with the expectations.

Cladera [17] published a study focused on university students' multifactor assessment of the quality of teaching. Together with the student questionnaire, they proposed to use the so-called importance–performance analysis (IPA). IPA measures the differences between how students consider a particular quality attribute to be important and how positively they perceive it using the average or median. The coordinates of each attribute correspond to one of four quadrants: 1. high importance and high performance, 2. low importance and high performance, 3. low importance and low performance, and 4. high importance and low performance. The relevant quadrant then helps determine which activities or processes should be emphasized when making changes. Attributes located in quadrants 2 and 3 can be considered redundant because they are not crucial from the student's point of view. In addition, the attributes in quadrant 2 (low priority and high-performance pressure) signal possible student overload. Attributes in quadrant 1 are perceived as necessary and meet student expectations at the same time. Therefore, they can be described as strengths of quality. Attributes in quadrant 4 have a high priority in the moment but little fulfilment by the teacher. They, therefore, provide enormous potential for improvement, which needs to be focused on [18]. The IPA technique is applied at the beginning of the semester when students formulate their expectations. At the end of the semester, there is a section on performance parameters. Both the teacher and the university management can then gain a visual representation of the crucial aspects of teaching as perceived by the students, which reveals the less satisfactory teaching features requiring improvement.

Cladera [17] found that the most important aspects of teaching include teacher enthusiasm, course organization, teaching and learning materials, student assessment methods

and the feedback provided, student interest, lecturer knowledge, the usefulness of tasks, and interaction in teaching. When monitoring performance, the aspects that are the essential attributes could be ranked as follows: students consider the teacher's relationship with them, organization (well-prepared teaching materials, availability, and consistency of goals), tasks, exams, enthusiasm, learning, and interaction. The main shortcomings mentioned by the students include the learning itself and the teacher's enthusiasm. These attributes, with high perceived importance but low performance, are a source of student dissatisfaction and represent a critical area for improving the quality of teaching [17].

Teaching strategies used by academics in lectures, exercises, and seminars undoubtedly influence the quality of higher education. They are, among other things, linked to educational resources of a printed or digital nature. Sikorova et al. [3] published a more extensive study of the use of professional resources in the university environment.

Although interest in electronic and online resources has come to the fore in recent decades, college students still strongly prefer printed materials when learning [19]. The authors of an extensive survey of students from 21 countries came to the same conclusion. A questionnaire survey of more than 10,000 students states that 80% of students prefer learning from printed materials. The main reasons include a better focus on learning and a more prolonged fixation on knowledge [20]. The research did not confirm the connection with either the cultural or socioeconomic differences of the students. Sikorova's [21] longitudinal study of more than a hundred university students in teaching disciplines also concluded that students prefer printed sources to electronic ones. The frequency of use of different types of resources in the observed five-year period did not differ either, except for the use of students' own notes from lectures and seminars in learning, which decreased slightly [21]. So far, isolated surveys show how online teaching has been able to influence the learning habits of the learning community. This may gradually be reflected by students' preference for the format of teaching materials. Rosli et al. [22] conducted a short questionnaire survey of hundreds of Faculties of Defence and Technology students: 25% of the students strongly agreed with the statement that multimedia-based teaching is more effective, 60% agreed, and only 1% disagreed. However, the results do not affect whether these preferences occurred only in connection with the change to distance learning methods. The sample of respondents was also focused on information technology.

Contemporary studies also point to the fact that the academic performance of university students is not directly related to the format of the resource they use [23–25]. On the contrary, students' learning outcomes correlate with the number of points achieved in the admissions process and gender rather than the choice of printed or electronic materials [26]. Roy, Inglis, and Alcock [25] analysed how reading comprehension differed between university students learning from the print format and students studying the same subject matter from digital multimedia materials.

Currently, no evidence suggests that the digital form of text has a positive effect on students' learning outcomes. Electronic materials do not significantly affect the comprehension of a text. If we talk about digital educational resources (and, therefore, also about ICT technologies supporting this format of teaching), in the parameter of the quality of higher education, it is more about the personal preferences of students than about better results in their learning.

## 2. Methodology

The research builds on a previous questionnaire survey on the quality of higher education. One of the areas the students identified as problematic was the diverse support for their learning strategies and learning outcomes. Therefore, we decided to examine the learning attribute in more depth concerning teaching strategies, methods, and forms, representing varying degrees of student support. The main research question is: "What teaching strategies do university students consider to be high-quality and supportive of their learning?"

We have selected in-depth interviews with chosen respondents, students of follow-up master's programs. We attempted to answer four specific research questions that are in line with the research areas and develop the main research questions:

(1)   What learning outcomes do teachers focus on during the student's learning trajectory?
(2)   Which teaching methods do teachers use in distinct forms of teaching?
(3)   What teaching methods help students in learning and why?
(4)   What is the role of ICT in the teaching and learning of university students?

Data collection took place using the focus groups method, with three groups of students. Due to the pandemic situation, in-depth interviews took place in the MS Teams online environment. We used the published experience of Schulze et al. [27] with leading focus groups in distance form. The interview with the first group, which included seven students, lasted 2 h 32 min, the second lasted 2 h 49 min (8 students), and the third lasted 1 h 43 min. This group was the least numerous, comprising four students. The student sample was designed to achieve maximum sample variability [28]. Therefore, it was possible to obtain representation from all university faculties, although not equally. A total of 19 students took part in the survey from the Faculty of Science, the Faculty of Education, the Faculty of Medicine, the Faculty of Arts, and the Faculty of Social Studies. Three researchers, each from a different faculty, took part in each interview. One was in the role of the main interviewer; the others monitored the focus groups and participated in the evaluation of each interview. The survey was conducted based on a constructed tool, which contained learning outcomes, teaching forms and methods, and the use of ICT technologies in teaching. All focus groups were recorded with the consent of the respondents. Subsequently, there was a literal transcription of the recordings. The coding was done using Atlas.ti, version 7. The first coding phase took place with cooperation between two researchers to achieve content agreement in open codes. Each researcher coded the part they were specialized in (e.g., learning outcomes). Thus, the data were processed using open and analytical coding [28]. Subsequently, axial coding was performed, and categories of codes of similar meaning were created, which were subsequently recombined. This procedure made it possible to identify core statements and ensure consistency in interpreting the data obtained.

## 3. Results

### 3.1. Students Prefer Learning Outcomes They Will Use in Practice

First, we attempted to identify which learning outcomes students consider important and why. Students across faculties and disciplines paid particular attention to practical experience and training. Respondent 1: "For me, practical teaching or practice is the most important." This opinion is mainly explained by the need to encounter and experience the situation, leading to the practical training of the necessary skills for future job performance and better acquisition and understanding of theoretical knowledge.

These internships are organized differently at individual faculties and fields of study. The consensus is that the theoretical-to-practical training ratio is inadequate, as there is little practical training during the study. Especially in some bachelor's fields, it is absent: "I missed internships during my bachelor's studies." (Respondent 2). Respondents appreciated every internship, albeit to a small extent.

Respondent 3 suggestively described another critical learning outcome: "I think that the most important is probably the motivation for further professional development . . . " Respondents describe motivation as a tool for learning, future practice, and overall development. They are motivated primarily by what they will use in future practice or what attracts their attention. Therefore, teachers should focus on these aspects in theoretical training.

Participants across disciplines are also aware of the importance of communication and relate it mainly to working life. They consider the ability to express their opinion to be an essential part of their communication skills. In addition, communication has frequently been mentioned regarding collaboration skills. Respondent 1: " . . . so that people, when

they leave the school, would be able to negotiate with their superiors, subordinates, and their colleagues."

Data analysis provided a compelling look at the knowledge and facts students acquire during teaching. On the one hand, they realize that the theoretical basis is an essential part of the study of every field; on the other hand, they agree that theoretical practice prevails in their study, which is not good. Information and knowledge are the primary realized learning outcomes. Respondent 4: "When I replayed my five-year studies, I concluded that the main part of those five years was devoted to teaching facts, gaining new knowledge and information." Respondent 4 is a student teacher, and the analysed data show that this is the typical learning outcome of their field of study. It no longer applies to the nursing and medical fields. Throughout the analysis of the data, the differences were evident. In the case of learning outcomes, these students were the most intensively trained, especially in the second half of the study.

We have identified other skills and abilities as essential learning outcomes, including flexibility, openness, critical thinking, and creativity. Participants explain their importance by the need to apply these skills in real life.

Data analysis allowed us to define the learning outcomes that should be taught within the curriculum. Students will appreciate a more thorough distinction between what is necessary and valuable for practice when selecting a curriculum. Teachers should pay attention to that regarding the learning outcomes. Academics should also focus on students' views, ways of thinking, and proposed solutions to problems, e.g., in model situations, and use them in learning so that students master the required learning outcomes. A teacher's openness to the topic, which can be reflected in their teaching, e.g., by the teacher's ability to accept students' ideas, think intensively about them, and clarify their content in the discussion, is appreciated. We noted that students strongly require their teacher to verify the expected learning outcomes' achievement regularly, making sure that the students understand the content of the curriculum. In the case of practical outputs, the teacher should provide regular feedback on their acquisition, since students need to fix their learning outcomes more intensively. However, the room for these activities is usually insufficient.

The analysis of the obtained data shows that students prefer the learning outcomes they believe will be used in practice. They concurrently understand that a theoretical basis is essential for practical readiness to pursue a future profession. However, they consider the theoretical and practical training ratio to be unbalanced.

*3.2. Characteristics of Teaching Forms Depend on Way Teacher Works*

The different choices of methods, room size, and the number of students usually distinguish the form of lectures, seminars, or exercises. Students do not find a difference between a seminar and an exercise at some faculties, or they confuse these forms. In some subjects, the differences between the lecture and the seminar are blurred depending on the teacher's motivation, amount and nature of the curriculum, or the number of students. Some students explain different conceptions of teaching forms during bachelor's and master's studies. As can be read in related methodological studies, e.g., in [29], the inspiration of good practices can be seen. Master's studies require and enable greater individualization. It also depends on whether the same teacher leads the lecture and the seminar. If the teacher is the same, the forms intertwine or are not distinguished. The teacher expresses the obligation to attend lectures by an attendance check. If it is not performed regularly or at least randomly, student attendance is limited to the most motivated.

Students characterize the lectures as monologues, as they focus on passing on the facts. Respondent 1: "Our lecture is purely monologued, with the lecturer sometimes asking something, but it is more or less about a monologue with a presentation." Teachers rely on presentations that are shown to students. During the presentation, some teachers ask students questions and lead discussions. This traditional form of the lecture is passive.

Students not only reject it but also sometimes welcome it, since the lecture provides less accessible information. Some lecturers rely on communication with students, interaction, and work in groups, even with a large number of students. This is perceived as a more effective procedure than a traditional lecture monologue. Respondent 5: "Active involvement of students in class is important. It leads to better memorization, the interconnection of the learning curriculum." More room for discussion opens when there are fewer students in the class or the lecture is interspersed with interaction with students, e.g., in the form of a reflection on reading.

Seminars that are smaller in number than lectures are described as rather dialogical and practically focused. They are also used to deepen theoretical knowledge or have their "lecture part". They are perceived as more effective than the lectures mainly due to the opportunity to communicate with the teacher, be more active, and cooperate with classmates. Presentations by the teacher or by students who independently study the topic and mediate it to others, helping them master it, are frequently used in seminars. Respondent 1: "The seminar usually takes place so that the students usually have presentations, or the presentations are made by the teachers, with more dialogue with the students." In these activities, work in pairs or groups is used, frequently with written output.

Exercises are the most practical and emphasize individualization. E.g., in the medical field, they are characterized by personal contact with patients, their causes, and examination methods. At school, they begin with a summary of theory, followed by calculations and practical skills development in the laboratory.

### 3.3. Students Appreciate Teaching Methods That Lead to Their Active Learning

Students appreciate teaching that requires regular preparation. However, the teacher's demands must be commensurate with the time prospects of the students. Although students consider homework before the lecture as an opportunity to perceive it more effectively, most teachers do not provide any materials in advance. If so, students lack sufficient motivation to self-study. Some keep notes from interactive lessons, review them, and consolidate the curriculum individually. Brief forms of the literature, such as textbooks and the structure of the lectures and seminars corresponding to them, are also appreciated as they allow students to acquire a good bearing on the subject. Some students prefer listening to lectures from recordings. Others pragmatically sacrifice activating teaching methods, favouring clearly and comprehensively created presentations that can be understood and reproduced during the exam. Others prefer to listen or write rather than communicate, as they may not be prepared for these methods.

Students expect a greater degree of transmissibility in lectures. They perceive it as a traditional way of passing on knowledge. Sometimes they notice reasons for the prevailing monologue in students' fears of asking questions when they do not understand, even though the teacher would be willing to answer them. Although the monologue sometimes prevailed in other forms, it was not always perceived negatively, but as a method that teachers can use to share their practical experiences with students (travel experiences, knowledge of the environment, and research results). This was true even if the memorization of the facts—except for stories—was low due to passive reception, especially in terms of long-term memorization supported by active involvement. Gradual involvement in activities allows students to become used to these methods and feel more comfortable utilizing them than when the teacher forces them to do so abruptly. Respondent 6: "I prefer activating methods, that is, a discussion allowing us to form our own opinion and defend it, use critical thinking." However, calling students up to the blackboard, for example, which most students do not like and only in exceptional cases consider beneficial (as it allows them to step outside of their comfort zone or have more direct experience with the exercise), is sometimes considered an activating method. Exercise tasks that are also balanced with the monologue part are welcomed. The development of presentation skills is also paramount. Students estimated the optimal share of activating methods in teaching to be 30% to 50%. It is also essential to understand the risk of the overuse of these

methods, which leads to overwhelming and demotivating students. When preparing for an internship, the teacher must leave the students with adequate space for independent work, as the students are responsible for the result.

Students consider seminars and exercises as a suitable form for using activating methods. Data analysis showed the main advantages of their usage. In particular, it can spark engagement in a problem, stimulate and develop interest, create space for cooperation, and maintain attention. Students usually consider dialogue, discussion, problem-solving and model situations, role-playing and dramatization, research and heuristic methods, critical thinking methods, various case studies, and project and group learning to be effective methods. In addition, working in pairs and groups strengthens social skills and allows for mastering more complex topics. Students understand dialogue as a method for communicating with the teacher and other students, developing critical thinking that supports the ability to quickly and effectively familiarize themselves with new problems. Dialogue helps to formulate ideas, defend one's own opinions, learn to ask questions, and react flexibly to such questions. It leads to deeper thinking and better memorization of topics. The model situation allows students to take on a different role and pushes their limits. Such experiential learning leads to better memorization and, thus, preparation for similar situations. Case studies are also directly linked to the practice, as they intertwine theory with practice. For example, in the form of dialogue, students attempt to uncover the essence of the problem and propose the right solution. Data analysis has shown that students value the methods that lead to their active learning.

### 3.4. Digital Technologies in Teaching

Students considered combining ICT technologies and social networks with distance learning a new experience. Online teaching has brought the challenge of technical hurdles as, according to some, online environments distort communication. At the same time, however, they reduce studying time and financial costs. During the pandemic, the transition to technology teaching revealed opportunities for greater access to digitized content (Kramerius), which students would welcome as a permanent service. The online mode has contributed to the development of hitherto underused learning methods, such as working in pairs and groups, including a new level of cheating. Respondent 7: "Many teachers started to get involved and look for alternative solutions that no one had anticipated until then because most teachers mainly relied on what they said in those classes." Online teaching demands commitment and engagement, since it is possible to pretend both physical presence and mental contact during teaching.

Students generally do not doubt that ICT belongs in teaching. Many of them consider its use motivating, as it leads to a greater variety of teaching methods and better fixation of the curriculum. According to others, ICT is not automatically a guarantee of higher-quality teaching. Overusing one method (presentation) can even weaken students' interest. Fatigue caused by many hours of online learning combined with ICT in everyday life should also be considered. The use of ICT is necessary for some subjects and serves to diversify and enrich the teaching of others. Respondent 7: "So we got various quizzes, questionnaires, and materials of all kinds, which was very beneficial." The need to communicate with a technologically excluded part of society and the danger of isolation from the natural world and nature itself were also mentioned. Respondent 8: "I would take away the tempo because art documents society, and if we want to do it, we want to keep it and, in some way, serve that society as artists, then we have to go into the field, and we must not become a generation that will only understand to this [ICT world]. We have to understand both human and animal things."

Some respondents see a problem in the fact that various technologies or computer applications in teaching are only talked about, without students learning to use them actively. For some, ICT means mainly the use of presentations, audio-visual content, and practice quizzes, i.e., technologies they have already experienced in teaching. ICT should not limit students' activity, and it should be used methodologically correctly. The advantage

of teaching using ICT arises in comparison with much less attractive and insufficiently inspiring previous teaching, which works only with textbook texts and pictures. Thanks to ICT, the opportunity to quickly search for information and use suggestions from all over the world is appreciated, as well as the opportunity to organize knowledge with the help of applications such as the OrgPad mind map. The idea that ICT disrupts learning was rejected because there are many other ways to procrastinate. The use of ICT is essential for the future of learning, which will be much more closely linked to it. It also makes knowledge accessible without barriers.

PowerPoint presentations in lectures are perceived as functional because they visualize the content, summarize what is already known, and add new information. Visual content tends to be more illustrative than the interpretation that does not work with them, and they are better remembered. However, the effect depends on its quality, a reasonable length, and the amount of information contained. TED lectures were given as an example of mastered presentations. It is considered perfect if a lecture is not based only on a presentation but uses more varied methods. Respondent 9: "When it's the TED version, it's the pictures behind me, one sentence, some quote I can develop, it's perfect. I focus on the person because there is the spotlight, or the presentation contains everything essential . . . ."

There are also questions about the actual effectiveness of presentations. A teacher's instructions on participating in a presentation lesson play an important role. Some students follow the presentation and the teacher's commentary; however, it is difficult for them to record both simultaneously. Others focus exclusively on the written capture of the projected presentation, if the teacher does not provide it to the students. Neither type of student can fully concentrate and provide feedback to the teacher. Teaching based on presentations does not develop the communication strategies that are needed, e.g., in pedagogical or psychological practice. The use of presentations in online teaching is even more demanding and leads to earlier fatigue. Only some students use the provided presentations as a teaching text. Others rely on videos, training applications (vocabulary), books, e-books, or scanned materials. Podcasts were also mentioned in connection with the strengthening of listening skills. Students search for additional information on the Internet. We were surprised by how few students learn from their notes.

*3.5. Distance Learning Has Strengthened Relations between Students and Teachers but Worsened Relations between Students*

Some students lost motivation to interact and communicate with others during the transition to teaching through ICT caused by the pandemic. The reason was the absence of a physical dimension of a mutual contact, which was frequently limited to writing and sending photos on social networks. On the other hand, the situation has contributed to developing students' ability to work together to solve assigned tasks and help and advise each other. However, a significant number of respondents agreed with the decline in student relations and the increase in the value of direct face-to-face contact. Michaela explained: "Face-to-face contact cannot be substituted by seeing ourselves here via a computer now; on the other hand, it's a great possibility that we can meet worldwide."

There was also agreement that communication between students and teachers improved compared to in-person teaching. ICT has made it easier and more economical for the benefit of better time management of both teachers and students, e.g., in online consultations; it has led teachers to be more interested in students' views and needs and to promote a collegial approach. A teacher's presence on the screen and their ability to respond to students' facial expressions encouraged the involvement of the students, since it is usually a step out of their comfort zone to be in front of others. On the other hand, other participants mentioned a decrease in the number of students involved in teaching. Uncertainty arose mainly in connection with exams, where students lacked direct engagement with their teacher, but, at the same time, teachers appreciated the students who were active during online teaching. Our data showed a change in the perception of mutual relations, and this important phenomenon should be the subject of further research.

## 4. Discussion

Focusing on the students' outcomes, the achieved results showed that students prefer the teaching and learning outcomes associated with the use in future practice. However, related to [10,11], the quality of teaching regarding its understanding has not been particularly discussed yet. Although, according to [7], teachers and their teaching strategies are the main factors that determine students' learning outcomes.

Regarding the teaching forms analysis, these forms have depended on the teacher's style rather than on the declared description in the curriculum. Cladera [17] found that the most important aspects of teaching include teacher enthusiasm, course organization, teaching and learning materials, student assessment methods and the feedback provided, student interest, lecturer knowledge, the usefulness of tasks, and interaction in teaching. As can be read in related methodological studies, e.g., in [29], the inspiration of good practices can be seen. Master's studies require and enable greater individualization, which also depend on whether the same teacher leads the lecture and the seminar. If the teacher is the same, the forms intertwine or are not distinguished.

ICT in teaching makes sense and positively impacts students' learning, but the effect depends on how ICT is used. Working with ICT technologies in distance learning has brought teachers and students closer together. Roy, Inglis, and Alcock [25] analysed how reading comprehension differed between university students learning from the print format and students studying the same subject matter from digital multimedia materials. Related to this analysis [25], the differences between the groups were not statistically significant, and students learning from the printed text achieved an even better score throughout the experiment.

Three essential topics have been identified: students' learning outcomes, teaching forms, and ICT methods. In addition, in the frame of the proposed and realized qualitative analysis, causality due to the transition to online teaching can be seen with regard to the mutual relations between students and teachers.

The possibilities of consideration for further research can be based on teaching methods regarding active students' learning. The preferred learning outcomes can be clarified by the approaches to their achievement within the individual methods. This topic also requires clarifying the proportional representation of theoretical and practical training in higher education.

The limits of the realized study can appear in the particular focus on the local character. However, the recommendations can be suitable for implementation in similarly based university types, in the case of the presented approaches.

## 5. Conclusions

The quality of university teaching and ways of learning by students is a current topic not only in Czechia. Research on this issue can focus on many areas. We have selected three primary topics for our research. First, we examined the outcomes that students prefer in teaching and learning. It turned out that students understand the need for a theoretical basis but mostly prefer the skills and abilities that they will use in their future profession. However, this finding conflicts with the composition of studies in most of the fields represented in our sample.

Research has revealed that most studies involve the transfer of theoretical facts and knowledge related to the field. The ratio of practical training is unbalanced and insufficient, except for medicine and nursing. In teaching forms, there is an overlap between the course and the organization of lectures and seminars, mainly depending on the teacher's concept. Students predominantly prefer seminars with smaller groups and the possibility to use activating methods. Students' learning is facilitated if the lectures have a clear structure and are supported by short learning materials—textbooks with a structure identical to the structure of the lectures. Exercises are appreciated mainly due to the possibility of practical training in activities, which students will use in their future profession. Monologue teaching methods predominate in teaching. However, students are helped by methods that lead

to their active learning. These are mainly the heuristic and activation methods. Students value methods that help shape their thinking and develop critical thinking. We have found the most significant discrepancy between what students prefer and what is implemented in practice in the field of teaching methods. ICT technologies are a standard part of higher education. ICT's use ensures a variety in teaching and positively impacts students' learning. This also leads to a greater variety of teaching materials, and their effectiveness and the ways of incorporating them into teaching are essential.

**Author Contributions:** Conceptualization, M.V.; theoretical chapters, I.C. and B.B; methodology, M.V., research questions, M.V. and M.T.; procedures of qualitative research, M.V. and M.T.; collected data, M.V. and M.T.; coded responses, M.V. and M.T.; formal analysis, M.V. and M.T.; discussion, M.V. writing—original draft preparation, M.V., M.T., I.C and B.B.; writing—review and editing, M.V. All authors have read and agreed to the published version of the manuscript.

**Funding:** This research received no external funding.

**Institutional Review Board Statement:** Not applicable.

**Informed Consent Statement:** Informed consent was obtained from all subjects involved in the study.

**Data Availability Statement:** The personal data of the students have been anonymized.

**Acknowledgments:** The authors acknowledge the Centre for Educational Research for providing materials and know-how of the utilized methodological approaches.

**Conflicts of Interest:** The authors declare no conflict of interest.

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
