# Peer review of "Analysis of Quality Teaching and Learning from Perspective of University Students"

_education, doi:10.3390/educsci12110820_

Round 1

Reviewer 1 Report

The authors chose a completely original and promising line of research. But in its current embryonic state, the work cannot be published.

Small suggestions:

The abstract in the current edition contains too general phrases that are not of much interest to the reader.

"According to a consensus in the higher education environment, student feedback improves teaching and helps develop effective teaching strategies." - we can agree with this, but are there alternative methods for developing effective teaching strategies? If so, why did the authors still choose the direction they chose?

The introduction as a whole is very interesting and convincing, but some paragraphs are too long, which makes them difficult to understand (this, in my opinion, flaw can be traced throughout the text).

Significant remarks:

The work seriously lacks statistical justification as well as a description of the possibility of interpolating the results. We are curious to read students' opinions and authors' comments, but the scientific community would like to see more scientific methods.

Although the authors, to a certain extent (this aspect can also be improved), compare and contrast the results obtained with the studies cited, it is difficult to agree that the sample presented by the authors is representative to compare results of this study with more convincing samples.

In general, the article contains a promising theoretical part, a weak empirical part, and good conclusions about the weak empirical part. Obviously, the authors should supplement and restructure the empirical part so that this material becomes convincing and useful for the scientific community.

Author Response

Reviewer’s Recommendation

The abstract in the current edition contains too general phrases that are not of much interest to the reader.

Authors’ Response

The abstract has been revised and modified according to this recommendation.

Reviewer’s Recommendation

"According to a consensus in the higher education environment, student feedback improves teaching and helps develop effective teaching strategies." - we can agree with this, but are there alternative methods for developing effective teaching strategies? If so, why did the authors still choose the direction they chose?

Authors’ Response

Row 45 was extended regarding the kind recommendation about the consideration of alternative methods for developing effective teaching strategies.

Reviewer’s Recommendation

The introduction as a whole is very interesting and convincing, but some paragraphs are too long, which makes them difficult to understand (this, in my opinion, flaw can be traced throughout the text).

Authors’ Response

The section of the introduction has been revised and modified according to this recommendation.

Reviewer’s Recommendation

The work seriously lacks statistical justification as well as a description of the possibility of interpolating the results. We are curious to read students' opinions and authors' comments, but the scientific community would like to see more scientific methods.

Authors’ Response

The proposed and realized analysis was primarily based on the qualitative type of research. Therefore, the authors’ team did not a priori consider the quantitative type of research.

Reviewer’s Recommendation

Although the authors, to a certain extent (this aspect can also be improved), compare and contrast the results obtained with the studies cited, it is difficult to agree that the sample presented by the authors is representative to compare the results of this study with more convincing samples.

Authors’ Response

According to the extended section of the Discussion, the limits of the presented research are widely clarified regarding the focus on particular local aiming. The considered selection sample was primarily based on generally applied methods of option known in the qualitative research design, e.g. in (Cohen, 2018).

Reviewer’s Recommendation

In general, the article contains a promising theoretical part, a weak empirical part, and good conclusions about the weak empirical part. Obviously, the authors should supplement and restructure the empirical part so that this material becomes convincing and useful for the scientific community.

Authors’ Response

The section Discussion was added regarding the presented usefulness of achieved results regarding established methodological approaches, with the description of further research possibilities and also with limitations. (row 423)

Reviewer 2 Report

The article is current and useful for academia. For its scientific value to be of adequate quality, I recommend the following revisions:

Methodology

Cite the first two sentences, which study the deeper research in the article follows, line 141, 142.

Weakness of the research uneven sample. Was the sample drawn at random? It is not said.

It would be appropriate to add differentiation of teaching procedures (methods and forms) for bachelor's and master's studies based on examples of good practice or studies to the methodology.

Discussion

An important part of the discussion is not included in the article, the author(s) do not discuss their results with those of other studies.

Author Response

Reviewer’s Recommendation

The article is current and useful for academia. For its scientific value to be of adequate quality, I recommend the following revisions:

Authors’ Response

Our team would like to kindly thank you for review’s feedback and appropriate recommendations.

Reviewer’s Recommendation

Cite the first two sentences, which study the deeper research in the article follows, lines 141, 142.

Authors’ Response

According to source [25], Fig. 3 in Section 4.2, the achieved p-value was lower than .001 therefore the sentence was repaired on the correct formulation “were statically significant” instead of “were not statically significant” (typographical mistake caused during translation process). The source was complemented as [25]. (row 141)

Reviewer’s Recommendation

Weakness of the research uneven sample. Was the sample drawn at random? It is not said.

Authors’ Response

According to the extended section of the Discussion, the limits of the presented research are widely clarified regarding the focus on particular local aiming. The considered selection sample was primarily based on generally applied methods of stratificated option known in the qualitative research design, e.g. in (Cohen, 2018, pp. 154).

Reviewer’s Recommendation

It would be appropriate to add differentiation of teaching procedures (methods and forms) for bachelor's and master's studies based on examples of good practice or studies to the methodology.

Authors’ Response

Consideration of the differentiation was particularly extended in the form of related source [29], where the inspiration of good practice can be seen. (row 254)

Reviewer’s Recommendation

Discussion
An important part of the discussion is not included in the article, the author(s) do not discuss their results with those of other studies.

Authors’ Response

The section Discussion was added regarding the connection of achieved results with established methodological approaches, with a description of further research possibilities and also with limitations. (row 423)

Reviewer 3 Report

The paper presents the results of empirical research focused on the quality of teaching and learning methods from the perspective of master's students at one of the Czech universities. The manuscript is clear, relevant for the field and presented in a well-structured manner. The cited references are mostly recent publications and They are relevant too.   The references do not  include an excessive number of self-citations.

The manuscript is scientifically sound and the experimental design is appropriate. Data for the qualitative research were collected through in-depth  interviews; the primary research method was focus groups.  The manuscript’s results are reproducible based on the details given in the methods section. The data is interpreted appropriately and consistently throughout the manuscript. The conclusions are consistent with the evidence and arguments presented. The involvement of ICT in teaching makes sense and positively  impacts students' learning, but the effect depends on how they are used

Author Response

Reviewer’s Recommendations

The paper presents the results of empirical research focused on the quality of teaching and learning methods from the perspective of master's students at one of the Czech universities. The manuscript is clear, relevant for the field and presented in a well-structured manner. The cited references are mostly recent publications and They are relevant too.   The references do not include an excessive number of self-citations.

The manuscript is scientifically sound and the experimental design is appropriate. Data for the qualitative research were collected through in-depth interviews; the primary research method was focus groups.  The manuscript’s results are reproducible based on the details given in the methods section. The data is interpreted appropriately and consistently throughout the manuscript. The conclusions are consistent with the evidence and arguments presented. The involvement of ICT in teaching makes sense and positively  impacts students' learning, but the effect depends on how they are used

Autors’ Responses

Our team would like to kindly thank you for review’s feedback and appropriate recommendations.

Round 2

Reviewer 1 Report

All reviewer comments have been answered. The reviewer agrees with these answers. The authors consider an important issue and approach it with scientific methods, but that approach does not bring significant new scientific knowledge.

Reviewer 2 Report

The article is now of good quality, all my comments have been incorporated. I recommend for publication.